# Self-Supervised Vertical Federated Learning

**Timothy Castiglia**
Department of Computer Science
Rensselaer Polytechnic Institute
Troy, NY 12180
`castit@rpi.edu`

**Shiqiang Wang**
IBM Research
Thomas J. Watson Research Center
Yorktown Heights, NY
`wangshiq@us.ibm.com`

**Stacy Patterson**
Department of Computer Science
Rensselaer Polytechnic Institute
Troy, NY 12180
`sep@cs.rpi.edu`

## Abstract

We consider a system where parties store vertically-partitioned data with a partially overlapping sample space, and a server stores labels on a subset of data samples. Supervised Vertical Federated Learning (VFL) algorithms are limited to training models using only overlapping labeled data, which can lead to poor model performance or bias. Self-supervised learning has been shown to be effective for training on unlabeled data, but the current methods do not generalize to the vertically-partitioned setting. We propose a novel extension of self-supervised learning to VFL (SS-VFL), where unlabeled data is used to train representation networks and labeled data is used to train a downstream prediction network. We present two SS-VFL algorithms: SS-VFL-I is a two-phase algorithm which requires only one round of communication, while SS-VFL-C adds communication rounds to improve model generalization. We show that both SS-VFL algorithms can achieve up to $2\times$ higher accuracy than supervised VFL when labeled data is scarce at a significantly reduced communication cost.

## 1 Introduction

Federated learning has become of recent interest to the research community [1, 2, 3] and has shown promise in several applications such as personalized healthcare, smart transportation, and predictive energy systems [4]. Federated learning algorithms allows a set of distributed parties to train a model without the need to directly share local data. *Vertical Federated Learning* (VFL) [5] algorithms are an important class of federated learning algorithms. In VFL scenarios, there are typically a small number of institutions that store data with the same sample space but different feature spaces. For example, a bank, a hospital, and an insurance firm may seek to predict a value of common interest, such as credit score. Each institution may have information on the same individuals but will hold different feature information (e.g., financial transactions, medical history, and vehicle accident reports). In VFL, each party typically trains a local *representation network* while the server combines the outputs of all parties' networks, known as *representations*, to train a final prediction model. We illustrate the VFL model in Figure 1a. VFL is in contrast to *horizontal federated learning* (HFL), where parties share a feature space but not the sample space. There has been much recent interest in VFL algorithms [6, 7, 8, 9, 10, 11] though there are still many open problems.

Workshop on Federated Learning: Recent Advances and New Challenges, in Conjunction with NeurIPS 2022 (FL-NeurIPS'22). This workshop does not have official proceedings and this paper is non-archival.

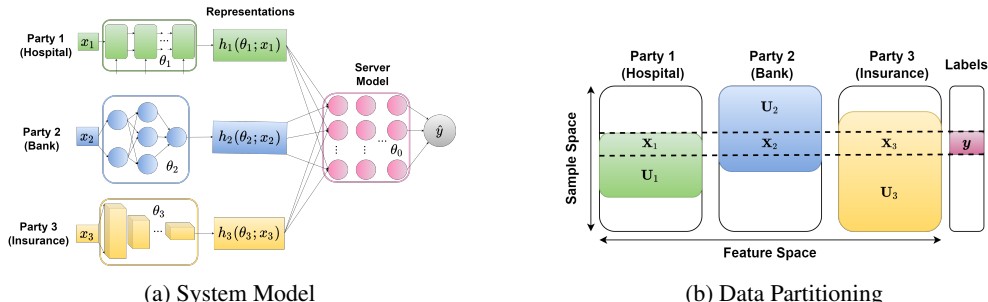

|  (a) System Model | (b) Data Partitioning |

Figure 1: $a$) Example VFL system model $\boldsymbol{\Theta}$. $b$) Example data partitioning. Overlapping labeled data for party $m$ is denoted by $\mathbf{X}_m$, and unlabeled data is denoted by $\mathbf{U}_m$.

VFL algorithms are typically *supervised*: models are updated based on how well their predictions compare to ground-truth *labels*. However, collecting labels can be a manual, time-consuming, and costly process. Often, only a small fraction of data is labeled. A major concern is that the labeled data does not represent the true underlying data distribution, and training on only labeled data can lead to model bias and poor model generalization [12]. Therefore, there is a need for incorporating *self-supervised* methods. Self-supervised learning (SSL) methods typically pre-train a model using unlabeled data followed by downstream supervised training [13]. Although there is a large body of work in SSL for HFL applications [14, 15, 16, 17], it has yet to be applied to the VFL setting.

Applying SSL to vertically-partitioned settings comes with a variety of unique challenges. SSL methods typically require full feature set access. However, each party in VFL only stores a subset of the total features for each sample. Sharing features in a VFL setting is often impossible due to data privacy or communication constraints. Additionally, in VFL settings, it is common that institutions have data for different individuals. This is illustrated in Figure 1b. These non-overlapping datasets can be very large, and it is necessary to integrate into the training process to avoid model bias and improve performance. Finally, many SSL algorithms assume parties have label access, which is not the case in VFL settings. Our work seeks to answer the following: *how can we conduct self-supervised learning on vertically-partitioned and potentially non-overlapping data?*

To answer this question, we propose two self-supervised VFL (SS-VFL) algorithms. Our first algorithm, SS-VFL Independent (SS-VFL-I), is a two-phase algorithm that requires only one round of communication between parties and the server, while the second, SS-VFL Coupled (SS-VFL-C), adds communication rounds to improve model generalization. In both algorithms, parties independently utilize contrastive learning for pre-training representation networks. Parties produce low-dimensional representations of local data without communication or access to labels. A major benefit of this approach is that parties' unlabeled datasets need not overlap. After unsupervised training, both algorithms are followed by downstream supervised training. SS-VFL-I proceeds by sending frozen representations to the server for final training, while SS-VFL-C updates representations during supervised training for stronger model performance at the cost of increased communication.

Our contributions are summarized as follows: 1) In Section 2, we formulate a self-supervised framework for training a model in a distributed system with a partially labeled vertically-partitioned dataset. 2) In Section 3, we propose SS-VFL-I, which improves performance over supervised VFL when labeled data is scarce, and has equivalent communication cost to only a single epoch of supervised VFL over the labeled data. 3) We propose SS-VFL-C, which trades communication savings for improving model performance by updating representation networks during downstream supervised training. SS-VFL-C has the same communication cost as supervised VFL. 4) In Section 4, we compare the SS-VFL algorithms with supervised VFL in a thorough set of experiments and show the SS-VFL algorithms can achieve up to $2\times$ the test accuracy of supervised VFL when labeled data is scarce, as well as significantly reduce communication cost to reach target accuracies.

**Related work:** Federated learning was first proposed by [18] with the goal of training a global model when participants store private data and the communication network faces high latency. Early works in Federated Learning have targeted the horizontally partitioned data scenario [1, 19, 20]. Self-supervised algorithms that utilize unlabeled data have been proposed in the past for the HFL setting [17, 14]. However, HFL algorithms rely on distributed gradient descent methods and share model parameter updates, while most VFL algorithms utilize distributed coordinate descent methods and share feature representations [7, 10, 11]. Thus, VFL algorithms are fundamentally different

from HFL algorithms. A few proposed VFL algorithms utilize unlabeled data. FedMVT, proposed by Kang et al. [21], uses a simple pseudo-labeling scheme that may be vulnerable to confirmation bias [22] and only applies to a two-party scenario. Cha et al. [23] propose a VFL algorithm that uses unsupervised learning with autoencoders as a pre-training step for supervised learning. Their work uses overcomplete autoencoders, which increase the dimensionality of raw data, causing the cost of communication to greatly increase over other VFL algorithms. Our proposed VFL algorithms use a method contrastive learning shown to avoid confirmation bias via data augmentation [22], apply to an arbitrary number of parties, and reduce communication cost.

## 2 Problem Formulation and Preliminaries

We present our problem formulation and provide background on related algorithms.

**Problem formulation:** We consider a network with $M$ parties and a server. Each party stores a set of features of a labeled dataset consisting of $N^l$ data samples. We let $\mathbf{X}_m$ denote the labeled data partition held by party $m$, and we let $x_m^i \in \mathbf{X}_m$ correspond to the data features that party $m$ stores on the $i$-th labeled data sample. The entire labeled data set is denoted by $\mathbf{X} = [\mathbf{X}_1, \ldots, \mathbf{X}_M]$, and $\mathbf{y}$ is the set of labels that corresponds to the data samples in $\mathbf{X}$. We assume that $\mathbf{y}$ is only present at the server. Each party $m$ also stores a set of features $\mathbf{U}_m$ corresponding to $N_m^u$ unlabeled data samples. We assume that $\mathbf{X}_m$ and $\mathbf{U}_m$ share the same feature dimension. Note, however, that we do not assume the parties necessarily store features for the same set of unlabeled samples. Thus, $N_m^u$ may not equal $N_j^u$ for parties $m \neq j$. Figure 1b illustrates the data partitioning among the parties.

The model consists of $M$ representation networks and a server prediction model. Each party $m$ stores its own representation network $h_m(\cdot)$, parameterized by $\theta_m$, that maps its local features to a representation space. The server's prediction model $f(\cdot)$, parameterized by $\theta_0$, combines representations from each party to make a prediction. An example of the model architecture is provided in Figure 1a. Let the model parameters be $\Theta = [\theta_0, \ldots, \theta_M]$, and let $\mathcal{D}$ be the distribution from which all data samples are drawn. The goal in training is to minimize the expected loss: $\mathbb{E}_{(x,y) \sim \mathcal{D}} f(\theta_0, h_1(\theta_1; x_1), \ldots, h_M(\theta_M; x_M); y)$. We next present existing methods for training over labeled and unlabeled data. We leverage these methods to develop our SS-VFL algorithms.

**Supervised VFL:** When data is labeled, one can use supervised VFL to train the model by minimizing the objective: $F(\Theta; \mathbf{X}, \mathbf{y}) \coloneqq \frac{1}{N^l} \sum_{i=1}^{N^l} f(\theta_0, h_1(\theta_1; x_1^i), \ldots, h_M(\theta_M; x_M^i); y^i)$. We provide pseudocode for supervised VFL [7, 24] in Appendix A. To summarize, each round starts with the parties agreeing on a randomly sampled mini-batch $\mathcal{B}^t$ from the labeled dataset. Each party inputs its corresponding features $\mathbf{X}_m^{\mathcal{B}^t}$ into its local representation network $h_m(\cdot)$ and sends the resulting representations to the server. The server then updates its model: $\theta_0^{t+1} = \theta_0^t - \eta^t \nabla_0 F_{\mathcal{B}}(\Phi^t; \mathbf{y}^{\mathcal{B}^t})$, where $\Phi^t$ is the set of all embeddings for $\mathcal{B}^t$, $\mathbf{y}^{\mathcal{B}}$ are the labels for batch $\mathcal{B}^t$, and $\eta^t$ is the step size at iteration $t$. Next, the server computes the partial derivative $\nabla_{h_m(\theta_m^t; \mathbf{X}_m^{\mathcal{B}^t})} F_{\mathcal{B}}(\Phi^t; \mathbf{y}^{\mathcal{B}^t})$ for all $m \in \mathcal{M}$. The server sends each partial derivative to its respective party. Each party then can compute the gradient update: $\nabla_m F_{\mathcal{B}}(\Phi^t; \mathbf{y}^{\mathcal{B}^t}) = h_m(\theta_m^t; \mathbf{X}_m^{\mathcal{B}^t})^\top \nabla_{h_m(\theta_m^t; \mathbf{X}_m^{\mathcal{B}^t})} F_{\mathcal{B}}(\Phi^t; \mathbf{y}^{\mathcal{B}^t})$, and update its model: $\theta_m^{t+1} = \theta_m^t - \eta^t \nabla_m F_{\mathcal{B}}(\Phi^t; \mathbf{y}^{\mathcal{B}^t})$.

**Unsupervised representation learning:** Contrastive learning is a centralized SSL solution to training with unlabeled data. In contrastive learning, it is assumed that for each data sample, there is access to a set of positive samples that share the same underlying structure as the original sample, as well as a set of negative samples that have a different underlying structure. In practice, these pairs are typically generated by creating data augmentations [25, 26, 27, 28]. The goal of contrastive learning is to train a representation network that maps data to a representation space where positive pairs are "close" to each other, and negative samples are "far" apart. This leads to representations clustered by their underlying features, making downstream supervised learning a simpler task.

For each class $c$, we let $D^c$ be the probability distribution over $\mathcal{X}$, where $\mathcal{X}$ is the set of all possible data points. $D^c(u)$ captures how relevant a sample $u$ is for a class $c$. Let $u$ and $u^+$ be data samples chosen i.i.d. from the same class, and let $u^-$ be a data sample chosen i.i.d. from a random class. We define $D^+(u, u^+)$ as the distribution of positive pairs, and let $D^-(u^-)$ be the distribution of negative samples. We define a similarity measure $\text{sim}(z_1, z_2) \coloneqq \exp(z_1^\top z_2 / \tau)$ where $\tau$ is a tunable temperature parameter. Contrastive learning trains a representation function $h(\cdot)$ on an unlabeled

---

**Algorithm 1** Communication-Efficient Self-Supervised Vertical Federated Learning

---

1: **Initialize:** $\theta_m^0$ for all parties $m$ and server model $\theta_0^0$
2: **for** $m \leftarrow 1, \ldots, M$ in parallel **do**
3:    $\theta_m^{T_{\text{un}}} \leftarrow \text{LOCALCL}(\theta_m^0, \mathbf{U}_m, T_{\text{un}})$
4:    Send representations $h_m(\theta_m^{T_{\text{un}}}; \mathbf{X}_m)$ to server
5: **end for**
6: **for** $t \leftarrow 0, \ldots, T_{\text{sup}} - 1$ **do**
7:    Randomly sample labeled mini-batch: $\mathcal{B}^t \in \{\mathbf{X}, \mathbf{y}\}$
8:    $\Phi^t \leftarrow \{\theta_0, h_1(\theta_1^t; \mathbf{X}_1^{\mathcal{B}^t}), \ldots, h_M(\theta_M^t; \mathbf{X}_M^{\mathcal{B}^t})\}$
9:    $\theta_0^{t+1} = \theta_0^t - \eta^t \nabla_0 F_{\mathcal{B}}(\Phi^t; \mathbf{y}^{\mathcal{B}^t})$
10: **end for**

---

dataset $\mathbf{U}$ by minimizing the objective [26]:

$$L(\theta; \mathbf{U}) \coloneqq \mathbb{E}_{\substack{(u, u^+) \sim D^+ \\ [u_i^-]_{i=1}^K \sim (D^-)^K}} \left[ -\ln \frac{\text{sim}(h(u), h(u^+))}{\text{sim}(h(u), h(u^+)) + \sum_i \text{sim}(h(u), h(u_i^-))} \right] \tag{1}$$

where $K$ is the number of negative samples. The numerator in (1) is maximized when the cosine similarity of positive pair representations is largest, while the denominator is minimized when the cosine similarity of negative samples is smallest. Contrastive learning has been proven to reduce the sample complexity of downstream supervised tasks [29].

## 3  Self-Supervised Vertical Federated Learning

We now present our algorithms for VFL with both labeled and unlabeled data. In both SS-VFL algorithms, each party $m$ independently runs contrastive learning to train its representation network. Similar to centralized contrastive learning, each party $m$ has a local class probability distribution $D_m^c(u)$ over $\mathcal{X}$ that captures the relevancy of a sample $u$ to a class $c$. The goal of each party is to minimize (1) for its local class distribution $D_m^c$. A party approximately minimizes (1) by randomly selecting a mini-batch $\mathcal{B}$ of samples, and updating its model using the gradient of the following:

$$L_{\mathcal{B}}(\theta_m; \mathbf{U}_m) \coloneqq \frac{1}{B} \sum_{u, u^+ \in \mathcal{B}} \left[ -\ln \frac{\text{sim}(h_m(u), h_m(u^+))}{\sum_{i \in \mathcal{B} \setminus \{u\}} \text{sim}(h_m(u), h_m(u^i))} \right] \tag{2}$$

The formal pseudocode of LocalCL is provided in Appendix A.

**SS-VFL-I:**  We now introduce SS-VFL-Independent (SS-VFL-I), a natural extension of contrastive learning to VFL. The pseudocode for SS-VFL-I is presented in Algorithm 1. At the start of training, the parties independently perform LocalCL to train their representation networks for $T_{\text{un}}$ iterations. Then, each party computes the representations for labeled data and sends these representations to the server. The server trains its prediction model on these representations without communication. SS-VFL-I only requires sending representations for all labeled data once; its communication is equivalent to a single epoch of supervised VFL, which can be immensely beneficial in situations where bandwidth is limited or costly. SS-VFL-I also provides inherent label privacy. It has been shown that supervised VFL can potentially leak label information through the sharing of partial derivatives [30, 31]. In SS-VFL-I, the server never communicates partial derivatives with the parties.

**SS-VFL-C:**  Next, we present SS-VFL Coupled (SS-VFL-C), an SS-VFL algorithm that improves representation networks during downstream supervised training while maintaining the same communication cost as supervised VFL. SS-VFL-C trades the communication savings of SS-VFL-I in order to update representation networks, improving downstream supervised model performance. The pseudocode for SS-VFL-C is presented in Algorithm 2. Just as in SS-VFL-I, each party uses local contrastive learning to train its representation network for $T_{\text{un}}$ iterations. However, the representation networks are not frozen at this point. In the second stage of SS-VFL-C, the same procedure as supervised VFL is followed. The parties share representations with the server, which trains a downstream prediction model. Then the server shares partial derivatives with the parties, and the parties update their representation networks. SS-VFL-C has the same communication cost as supervised VFL, but with the benefit of representation networks bein pre-trained using unlabeled data.

**Algorithm 2** Self-Supervised Vertical Federated Learning with Representation Updates

1: **Initialize:** $\theta_m^0$ for all parties $m$ and server model $\theta_0^0$
2: **for** $m \leftarrow 1, \ldots, M$ in parallel **do**
3: $\quad \theta_m^{T_{\text{un}}} \leftarrow \text{LOCALCL}(\theta_m^0, \mathbf{U}_m, T_{\text{un}})$
4: **end for**
5: **for** $t \leftarrow 0, \ldots, T_{\text{sup}} - 1$ **do**
6: $\quad$ Parties choose randomly sampled mini-batch: $\mathcal{B}^t$.
7: $\quad$ **for** $m \leftarrow 1, \ldots, M$ in parallel **do**
8: $\quad\quad$ Party sends representation $h_m(\theta_m^t; \mathbf{X}_m^{\mathcal{B}^t})$ to server
9: $\quad$ **end for**
10: $\quad \Phi^t \leftarrow \{\theta_0, h_1(\theta_1^t; \mathbf{X}_1^{\mathcal{B}^t}), \ldots, h_M(\theta_M^t; \mathbf{X}_M^{\mathcal{B}^t})\}$
11: $\quad \theta_0^{t+1} = \theta_0^t - \eta^t \nabla_0 F_{\mathcal{B}}(\Phi^t; \mathbf{y}^{\mathcal{B}^t})$
12: $\quad$ Server sends $\nabla_{h_m(\theta_m^t; \mathbf{X}_m^{\mathcal{B}^t})} F_{\mathcal{B}}(\Phi^t; \mathbf{y}^{\mathcal{B}^t})$ to each party $m$
13: $\quad$ **for** $m \leftarrow 1, \ldots, M$ in parallel **do**
14: $\quad\quad \nabla_m F_{\mathcal{B}}(\Phi^t; \mathbf{y}^{\mathcal{B}^t}) = h_m(\theta_m^t; \mathbf{X}_m^{\mathcal{B}^t})^\top \nabla_{h_m(\theta_m^t; \mathbf{X}_m^{\mathcal{B}^t})} F_{\mathcal{B}}(\Phi^t; \mathbf{y}^{\mathcal{B}^t})$
15: $\quad\quad \theta_m^{t+1} = \theta_m^t - \eta^t \nabla_m F_{\mathcal{B}}(\Phi^t; \mathbf{y}^{\mathcal{B}^t})$
16: $\quad$ **end for**
17: **end for**

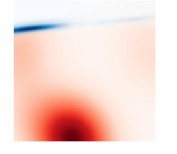
(a) CL with all features

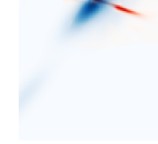
(b) CL with partial features

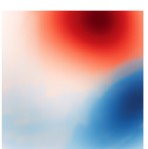
(c) After supervised training

Figure 2: Kernel density expectation maps of ImageNet representations reduced to two dimensions.

**LocalCL:** *When is LocalCL sufficient to create representations that can be easily classified during downstream supervised training?* In Figure 2, we give an example of when LocalCL fails to produce representations that are well-separable. We compare representations generated by running centralized contrastive learning with a full feature set and representations generated by running LocalCL with a partial feature set. Here, we plot a kernel density map of representations from the ImageNet dataset after being reduced to two dimensions (using PCA) and normalized to the unit circle. We show representations from two classes marked in red and blue. Ideally, the representations should be separated into distinct clusters without any overlap. For LocalCL, we consider a case where only half of each image is available. In Figure 2a, when all features are available for constrastive learning, we can see that representations from the different classes are correctly mapped into separate clusters. In Figure 2b, we can see that LocalCL produces clusters that overlap. Since local contrastive learning only has access to a partial feature set, two label classes may be indistinguishable with the available features, making downstream supervised training more difficult. In Figure 2c, we show the representations of two different classes at a single party after running SS-VFL-C. For cases when LocalCL produces overlapping representations, SS-VFL-C can be run to refine the representations. Formally, the difference between local and centralized contrastive learning manifests in the difference between $D_m^c$, the relevance score of a sample based on locally available features and $D^c$, the relevance score based on all features. For example, if a party only stores the lower half of an image for a table, it may be difficult to discern if the image are of the legs of a table or a chair. In cases where $D^c$ and $D_m^c$ are similar, then SS-VFL-I can provide strong representations for downstream training without high communication cost. Otherwise, SS-VFL-C provides a means of updating representation networks during downstream training at the cost of additional communication.

## 4 Experiments

We now present experiments comparing SS-VFL-I and SS-VFL-C with supervised VFL. The parties train their models on two datasets: ModelNet10 [32] and ImageNet [33]. ModelNet10 is a set of 2D images of 3D CAD models from different camera views. For this dataset, there are 12 parties, each

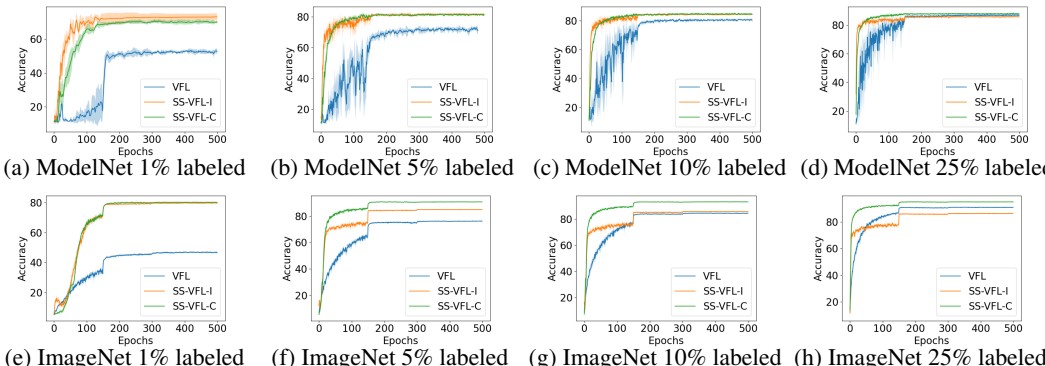

(a) ModelNet 1% labeled (b) ModelNet 5% labeled (c) ModelNet 10% labeled (d) ModelNet 25% labeled

(e) ImageNet 1% labeled (f) ImageNet 5% labeled (g) ImageNet 10% labeled (h) ImageNet 25% labeled

Figure 3: Top-5 test accuracy after running supervised VFL, SS-VFL-I and SS-VFL-C on the ModelNet10 and ImageNet100 datasets with $1\%$, $5\%$, $10\%$, and $25\%$ of the training dataset labeled. The solid lines are the mean of 5 runs, while the shaded region represents the standard deviation.

Table 1: Communication cost in MB of VFL, SS-VFL-I, and SS-VFL-C on the ImageNet100 and ModelNet10 datasets to reach a target test accuracy with different fractions of labeled data. For ImageNet100, we use top-5 accuracy, and for ModelNet10, we use top-1 accuracy. The value shown is the mean of 5 runs $\pm$ the standard deviation.

| Labeled Fraction | Target Accuracy | Communication Cost (MB) to reach target | | | | | |
| | | ImageNet100 dataset | | | ModelNet10 dataset | | |
| | | VFL | SS-VFL-I | SS-VFL-C | VFL | SS-VFL-I | SS-VFL-C |
|---|---|---|---|---|---|---|---|
| 1% | 70% | – | 1.24 | $143.63 \pm 9.40$ | – | 0.23 | $23.02 \pm 2.27$ |
| 5% | 75% | $1023.96 \pm 43.88$ | 6.19 | $153.53 \pm 2.48$ | $175.55 \pm 31.34$ | 1.17 | $37.03 \pm 8.17$ |
| 10% | 80% | $1857.25 \pm 0.00$ | 12.38 | $210.49 \pm 13.56$ | $225.47 \pm 52.77$ | 2.34 | $55.31 \pm 5.05$ |
| 25% | 85% | $3089.22 \pm 111.43$ | 30.95 | $396.21 \pm 49.53$ | $330.47 \pm 83.90$ | 5.86 | $104.30 \pm 8.61$ |

with one view of every CAD model. Each party trains ResNet18 as a representation model, while the server trains a fully connected layer. ImageNet is a set of images of different objects and animals. For our experiments, we choose a random set of 100 classes from ImageNet (ImageNet100). For ImageNet100, there are two parties, and each party stores half of each image. Each party uses ResNet50, while the server trains a fully connected layer. For SS-VFL-I and SS-VFL-C, parties run LocalCL for 200 epochs. Then, all algorithms train on the labeled data for 500 epochs. We consider cases where the training set of each dataset has only $1\%$, $5\%$, $10\%$, or $25\%$ of its data labeled. We run each algorithm on both datasets with each of these labeled fractions.

The results of the experiments are shown in Figure 3, where we plot the test accuracy for each dataset and labeled data fraction. The solid lines are the mean of 5 runs, while the shaded regions represent the standard deviation. For the ModelNet10 dataset, we can see at $1\%$ and $5\%$ labeled data, both SS-VFL algorithms outperform supervised VFL. Only at $10\%$ labeled data and more is supervised VFL able to reach similar accuracies to the SS-VFL algorithms. We can see that SS-VFL-I and SS-VFL-C perform similarly in all cases, indicating that LocalCL was able to produce well-separable representations during unsupervised training. For the ImageNet100 dataset, both SS-VFL algorithms perform similarly when $1\%$ of the data is labeled, reaching up to double the accuracy of supervised VFL. As the amount of labeled data increases, we can see that SS-VFL-C continues to reach higher test accuracy than the other two algorithms, providing the best model generalization by utilizing both labeled and unlabeled data to train party representation networks. In the case of ImageNet100, LocalCL has more difficulty distinguishing between similar classes. SS-VFL-C outperforms SS-VFL-I here by utilizing additional communication during downstream supervised training, allowing it to refine the representations and improve performance.

In Table 1, we show the communication cost between the parties and the server for supervised VFL, SS-VFL-I, and SS-VFL-C to reach a target test accuracy. We can see in Table 1 that SS-VFL-I has a much smaller communication cost than both other algorithms, regardless of the fraction of labeled data. SS-VFL-C, although requiring more communication, still reduces overall communication cost to reach target accuracies over supervised VFL in both datasets. For scenarios where labeled data is limited, both SS-VFL algorithms provide immense benefits in communication reduction.

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

# A  Algorithm Pseudocode

---

**Algorithm 3** Supervised Vertical Federated Learning

---

1: **Initialize:** $\theta_m^0$ for all parties $m$ and server model $\theta_0^0$
2: **for** $t \leftarrow 0, \ldots, T-1$ **do**
3:      Parties choose randomly sampled mini-batch: $\mathcal{B}^t$.
4:      **for** $m \leftarrow 1, \ldots, M$ in parallel **do**
5:          Party sends representation $h_m(\theta_m^t; \mathbf{X}_m^{\mathcal{B}^t})$ to server
6:      **end for**
7:      $\Phi^t \leftarrow \{\theta_0, h_1(\theta_1^t; \mathbf{X}_1^{\mathcal{B}^t}), \ldots, h_M(\theta_M^t; \mathbf{X}_M^{\mathcal{B}^t})\}$
8:      $\theta_0^{t+1} = \theta_0^t - \eta^t \nabla_0 F_{\mathcal{B}}(\Phi^t; \mathbf{y}^{\mathcal{B}^t})$
9:      Server sends $\nabla_{h_m(\theta_m^t; \mathbf{X}_m^{\mathcal{B}^t})} F_{\mathcal{B}}(\Phi^t; \mathbf{y}^{\mathcal{B}^t})$ to each party $m$
10:     **for** $m \leftarrow 1, \ldots, M$ in parallel **do**
11:        $\nabla_m F_{\mathcal{B}}(\Phi^t; \mathbf{y}^{\mathcal{B}^t}) = h_m(\theta_m^t; \mathbf{X}_m^{\mathcal{B}^t})^\top \nabla_{h_m(\theta_m^t; \mathbf{X}_m^{\mathcal{B}^t})} F_{\mathcal{B}}(\Phi^t; \mathbf{y}^{\mathcal{B}^t})$
12:        $\theta_m^{t+1} = \theta_m^t - \eta^t \nabla_m F_{\mathcal{B}}(\Phi^t; \mathbf{y}^{\mathcal{B}^t})$
13:     **end for**
14: **end for**

---

In Algorithm 3, we provide the pseudocode of Supervised Vertical Federated Learning, as described in Section 2.

---

**Algorithm 4** Local Contrastive Learning

---

1: **procedure** LOCALCL($\theta, \mathbf{U}_m, T$)
2:      **for** $t \leftarrow 0, \ldots, T-1$ **do**
3:          Randomly sample unlabeled mini-batch: $\mathcal{B}^t \in \mathbf{U}_m$
4:          $\theta^{t+1} = \theta^t - \eta^t \nabla L_{\mathcal{B}}(\theta^t; \mathcal{B}^t)$
5:      **end for**
6: **end procedure**

---

In Algorithm 4, we provide the pseudocode of local constrastive representation learning, as described in Section 3.

# B  Societal/Ethical impacts

With any machine learning task, there is the concern of the resulting model learning a bias that may discriminate against groups of people unfairly. Although utilizing SS-VFL can mitigate model bias by including unlabeled data, biased data collection can still lead to model bias. Additionally, a "protected" feature, such as race, may play a large role in the prediction model. Researchers and developers must be aware of the potential for bias in the datasets and take steps to ensure that these biases do not lead to discriminatory practices. Several techniques to correct for bias in datasets exist and can be applied to SS-VFL. For example, one can apply weighting to the data samples to correct for a known bias in the dataset.

