# OpenReview forum: "Self-Supervised Vertical Federated Learning"
_NeurIPS.cc/2022/Workshop/Federated_Learning — FL-NeurIPS 2022 Poster_

### Official Review · Reviewer_5DTN · 2022-10-16

This work proposes to combine self-supervised learning (SSL) with Vertical Federate Learning (VFL). It demonstrates two variations including SS-VFL-I and SS-VFL-C.

Novelty:
The novelty of the proposed work is limited. The proposed methods including SS-VFL-I and SS-VFL-C treat SSL as a warm-up. Thus, it is a natural extension of VFL with SSL. Furthermore, SS-VFL-I is only one particular case of SS-VFL-C. The paper should instead conduct ablation studies on the number of epochs in the second step.

Quality:
The quality of this work is fair. Figure 2 does not seem to be informative. It is hard tell Figure 2(b) produces overlapped features. The number of parties in the ImageNet100 dataset is small. They paper may instead use CIFAR100 and split into 8 different patches [1].

[1] GAL: Gradient Assisted Learning for Decentralized Multi-Organization Collaborations

Clarity:
The clarity of this work is good.

Significance:
The significance of this work is limited. A natural combination of SSL with VFL without theoretical analysis generally does not qualify top-venue conferences. There also exist many other ways of leveraging unlabeled data such as semi-supervised learning. The paper may conduct more ablation studies including the number of parties and the number of epochs.

Why does SS-VFL-I perform worse than VFL in Figure 3(h)?

---

### Official Review · Reviewer_RhQj · 2022-10-17
**interesting topic, limited novelty, lack of experiments**

This paper proposes to leverage self-supervised learning to train representation models to improve the performance of VFL when the labeled data is scarce. The paper presents two self-supervised VFL algorithms, SS-VFL-I and SS-VFL-C. The experiment results on two image datasets demonstrate that both algorithms can achieve much higher accuracy than supervised VFL when the labeled data is scarce. Meanwhile, SS-VFL-I and SS-VFL-C can achieve the same accuracy with supervised VFL at a significantly reduced communication cost.

Strengths:
1. The paper is well-written and easy to follow.
2. This paper studies self-supervised vertical federated learning, which is an interesting problem.

Weaknesses:
1. The novel of this paper is limited. The proposed algorithms SS-VFL-I and SS-VFL-C just simply adopt contrastive learning to train representation models of all parties. In each party, the trained local representation models only consider their own features and cannot capture the relationships between features owned by other parties. In VFL, it is more important to design a privacy-preserving approach to conduct global contrastive learning on all parties’ features.
2. There are many self-supervised techniques, for example, Generative, Contrastive, and Generative-Contrastive. This paper chooses contrastive learning to perform self-supervised learning in VFL, but does not explain why. It is better to explain why contrastive learning is more suitable for self-supervised learning in VFL.
3. The experiments are not quite comprehensive. Only two image datasets are used and only supervised VFL baseline is compared. It is better to conduct experiments on more datasets and compare SS-VFL-I/C with other self-supervised VFL works (e.g., Cha et al. [23]).

---

### Official Review · Reviewer_Bcpq · 2022-10-18

Summary:
	This paper proposes a VFL algorithm for dataset with small proportion labeled data. The proposed method first use contrastive learning on each client with the local data with partial feature, and then send all the extracted local features after contrastive learning to the server. Finally, the server trains server classifier based on the aggregated features.

Strength:
	The proposed algorithm can make use of data with partially overlapping samples and partially overlapping features, while the inference relies on all agents. The experiments showed that the proposed algorithm achieves better accuracy and greatly reduces communication when the proportion of the labeled data is small.

Problems:
1)	Whether contrastive learning can be done with partial features is unclear. There is an underlying key assumption for local contrastive learning, that the similarity between the partial features of samples from different classes is low.
However, this assumption implies that the classification can already be done with only clients’ partial features. Therefore, the comparison between stand-alone training (each client locally trains a model without any communication), and the proposed method (SS-VFL-I) is needed.

On the other hand, consider the example provided by the author in line 184 that the client features of a chair and a table are similar, it is unclear how SS-VFL-C can refine the local representation. For example, in the extreme example of digit classification on seven-segment display digits, and each feature can be the on-off status of one LED. For example, digit 1 is represent as (0110000), digit 2 is (1101101). Then each client cannot distinguish between the 10 digits with only one binary feature. So, in this example, Can the author elaborate more on how contrastive learning can be done?
2)	There is another missing piece of the experiment, that under the usual setting with sufficient labeled data, what will be the comparison between VFL and the proposed algorithm?
3)	I am a bit concerned on the fairness of the comparison in Figure 3 as the SS-VFL are pretrained with contrastive learning (200 epochs). Should VFL runs 200 or more epochs for fair comparison?
4)	There is a privacy concern that the extracted local features are aggregated by the server for server-side model update. In the other papers, the feature aggregation or exchanged can be encrypted or add other privacy protection

---

### Decision · Program_Chairs · 2022-10-20

Accept (Poster)